# DePLM: Denoising Protein Language Models for Property Optimization

**Zeyuan Wang**[1,2]    **Keyan Ding**[2]    **Ming Qin**[1,2]    **Xiaotong Li**[1,2]    **Xiang Zhuang**[1,2]
**Yu Zhao**[4]    **Jianhua Yao**[4]    **Qiang Zhang**[3,2]*    **Huajun Chen**[1,2]*
[1]College of Computer Science and Technology, Zhejiang University
[2]ZJU-Hangzhou Global Scientific and Technological Innovation Center
[3]The ZJU-UIUC Institute, International Campus, Zhejiang University
[4]Tencent AI Lab, Tencent
{yuanzew,dingkeyan,qinandming,3190104904}@zju.edu.cn
yu.zhao@tum.de    jianhua.yao@gmail.com
{zhuangxiang,qiang.zhang.cs,huajunsir}@zju.edu.cn

## Abstract

Protein optimization is a fundamental biological task aimed at enhancing the performance of proteins by modifying their sequences. Computational methods primarily rely on evolutionary information (EI) encoded by protein language models (PLMs) to predict fitness landscape for optimization. However, these methods suffer from a few limitations. (1) Evolutionary processes involve the simultaneous consideration of multiple functional properties, often overshadowing the specific property of interest. (2) Measurements of these properties tend to be tailored to experimental conditions, leading to reduced generalizability of trained models to novel proteins. To address these limitations, we introduce Denoising Protein Language Models (DePLM), a novel approach that refines the evolutionary information embodied in PLMs for improved protein optimization. Specifically, we conceptualize EI as comprising both property-relevant and irrelevant information, with the latter acting as "noise" for the optimization task at hand. Our approach involves denoising this EI in PLMs through a diffusion process conducted in the rank space of property values, thereby enhancing model generalization and ensuring dataset-agnostic learning. Extensive experimental results have demonstrated that DePLM not only surpasses the state-of-the-art in mutation effect prediction but also exhibits strong generalization capabilities for novel proteins.

## 1 Introduction

Proteins play vital roles in numerous biological processes, shaping their structure and function over billions of years of evolution. This evolutionary diversity presents significant opportunities for advancing fields such as drug discovery and materials science [24, 60]. However, the inherent properties of existing proteins, such as thermostability, often fall short of practical requirements in various scenarios. Consequently, researchers have dedicated themselves to optimizing proteins to enhance their properties of interest. Protein optimization is the task that involves modifying protein sequences and efficiently identifying well-performing proteins.

Traditional deep mutational scans (DMS) and directed evolution (DE) rely on expensive wet-lab experiments [15, 6, 58]. Recently, computational approaches that accurately model the relationship between proteins and their property fitness, often termed a "fitness landscape" [51], have become

---

*Corresponding author.

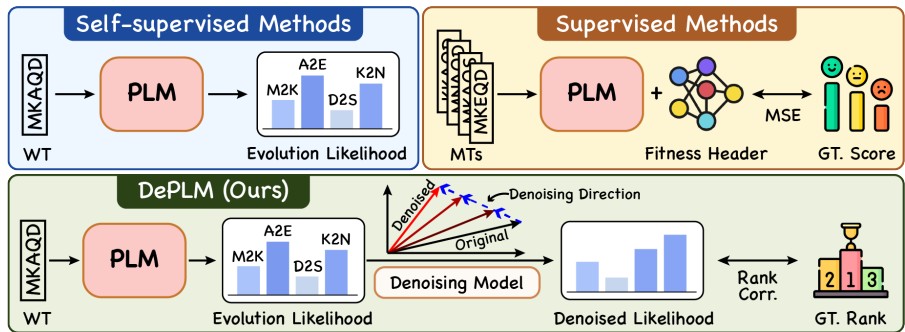

Figure 1: Comparison of fitness landscape prediction methods. WT: wildtype sequence. MTs: mutant sequences. A2B: Amino acid A in wildtype mutates to B. GT.: groundtruth. Corr.: Correlation.

crucial for efficient protein optimization. These approaches, powered by machine learning, enable rapid evaluation of mutation effects and are pivotal for efficient protein optimization.

One widely explored avenue involves leveraging the "Evolutionary Information" (EI), which can be instantiated by the likelihood of an amino acid appearing at a certain position of protein sequences, to infer the mutational effects [41, 49, 16]. This stems from the observation that as organisms evolve through natural selection, mutations that improve functional properties become more prevalent. Therefore, the likelihood of a mutation occurring is directly linked to its impact on biological function [20]. To compute the informative likelihoods of mutating one amino acid to another, the predominant methods involve protein language models (PLMs) trained on millions of protein sequences, which capture the EI in a self-supervised manner [50, 43, 39]. Thanks to their strong generalization capabilities, PLMs have been utilized to guide the artificial selection of beneficial mutations with notable efficacy [24].

With the advent of high-throughput experimentation, large-scale and diverse annotated datasets for DMS are becoming increasingly accessible [44]. Consequently, researchers have extended the use of self-supervised EI into supervised prediction settings [18, 17, 23, 21, 11, 65]. Specifically, they usually fine-tune PLMs on experimentally annotated datasets, with the objective of minimizing the disparities between predicted and experimental fitness values [8, 2, 46]. However, two critical aspects are often overlooked. Firstly, it fails to account for the removal of irrelevant EI. Evolution optimizes multiple properties simultaneously to meet survival needs, often overshadowing the optimization target of interest [35]. Therefore, conventional fine-tuning methods using the whole evolutionary information are suboptimal. Secondly, the prevalent learning objective incorporates dataset-specific information that is often overfitted to the training data at hand, hindering the model's ability to generalize toward new proteins. This limitation is significant since DMS experimental techniques often encounter constraints regarding their applicability across a wide range of proteins [58].

In this work, we introduce a novel **De**noising **P**rotein **L**anguage **M**odel (DePLM) tailored for protein fitness prediction. *The central concept revolves around perceiving the EI captured by PLMs as a blend of property-relevant and irrelevant information, with the latter akin to "noise" for the targeted property, necessitating its elimination.* To achieve this, drawing inspiration from denoising diffusion models that refine noisy inputs to generate desired outputs [55, 10, 29], we devise a rank-based forward process to extend the diffusion model for denoising EI, as illustrated in Figure 1. Specifically, we refine the likelihood of mutations provided by PLMs. To parameterize this framework, we initially extract protein representations considering both primary and tertiary structures. Subsequently, we utilize this representation to guide the denoising process. In pursuit of dataset-agnostic learning and robust model generalization, we conduct the diffusion process in the rank space of property values and replace the conventional objective of minimizing numerical errors with maximizing rank correlation. Extensive experiments have demonstrated that the introduced rank-based denoising process significantly improves the protein fitness prediction performance, and simultaneously maintains strong generalization ability for novel proteins. Our contributions can be summarized as follows:

- We introduce DePLM, a novel approach for refining evolutionary information captured by protein language models to predict mutation effects, effectively filtering out irrelevant information to improve mutation effect predictions.

- We design a rank-based forward process within the denoising diffusion framework, extending the diffusion process to the rank space of mutation likelihoods.
- We shift the learning objective from minimizing numerical errors to maximizing rank correlation, fostering dataset-agnostic learning and ensuring robust generalization.
- DePLM significantly outperforms state-of-the-art models for mutational effect prediction and demonstrates strong potential for optimizing unseen proteins.

## 2 Background and Related Works

### 2.1 Task Formulation and Evaluation

Protein optimization seeks to elucidate the impacts of sequence mutations on protein properties. Formally, given a widetype protein sequence $\boldsymbol{x}^{\mathrm{wt}} = [x_1^{\mathrm{wt}}, \cdots, x_n^{\mathrm{wt}}, \cdots, x_N^{\mathrm{wt}}]$ with $N$ amino acids, a mutation $\boldsymbol{\mu} = \{\mu_n : x_n^{\mathrm{wt}} \to x_n^{\mathrm{mt}}, n \in [\![1, N]\!]\}$ refers to the substitution of the amino acid $x_n^{\mathrm{wt}}$ at certain positions $n$ with another amino acid $x_n^{\mathrm{mt}}$. Note that a mutation can affect multiple positions simultaneously. The task of protein optimization is mathematically formulated as learning a function $\mathcal{F}_\theta$, parameterized by $\theta$: $\mathcal{F}_\theta(\boldsymbol{x}^{\mathrm{wt}}, \boldsymbol{\mu}) = y$, where $y$ denotes the impact of the mutation $\boldsymbol{\mu}$ on the wildtype sequence $\boldsymbol{x}^{\mathrm{wt}}$, i.e., the property value of the mutated protein sequence. With numerous $(\boldsymbol{\mu}, y)$ pairs, one can approximate the fitness landscape of the protein $\boldsymbol{x}^{\mathrm{wt}}$ for the target property.

To evaluate the consistency between predicted and ground-truth fitness landscape, one often uses Spearman's rank correlation coefficient, which prioritizes relative rankings over absolute values [44]. Specifically, given the set of predictions $Y = \{y\}$ and ground-truth $Y^* = \{y^*\}$, both sets are first converted to their respective ranks $R(Y)$ and $R(Y^*)$, then the coefficient $\rho$ is calculated as

$$\rho = \frac{\mathrm{cov}(R(Y), R(Y^*))}{\sigma_{R(Y)}\sigma_{R(Y^*)}}, \tag{1}$$

where $\mathrm{cov}(\cdot, \cdot)$ is the covariance of the ranked variables, $\sigma$ is the standard deviations of these ranks.

### 2.2 Related Works

**Self-supervised modeling of sequence mutation effects.** Evolutionary information provides valuable insights into how sequence mutations affect biological functions [14]. A straightforward approach to capture EI is through Multiple Sequence Alignments (MSAs). For example, the SIFT model [41] predicted the mutation effect by performing position-specific statistical analysis of aligned sequences. Riesselman et al. [49] and Frazer et al. [16] employed variational autoencoder trained on protein-specific MSAs to detect patterns of interaction among positions, achieving higher prediction performance. However, the MSA approach is limited by its protein-specific nature, rendering it less effective for proteins, such as orphan proteins, that lack sufficient homologous sequences.

Instead, many studies [2, 40, 50, 39, 4, 37] explored PLMs trained on evolutionary-scale data to capture EI, which can generalize beyond specific proteins. Further, several studies aim to merge PLMs with MSAs to harness the benefits of both approaches. Rao et al. [48] proposed MSA Transformer to apply language modeling to aligned sequences, and PoET [59] modeled the distribution over protein families rather than sequences. Tan et al. [56] suggested that mutation effects are related to their structural context, and Notin et al. [44] showed that the likelihood derived from structure-based design models [28, 9] can be complementary to those generated by PLMs. Despite the utility of EI, the information captured by PLMs is often entangled with multiple protein properties, leading to suboptimal optimization of the target property when used directly.

**Supervised modeling of fitness landscape.** The utility of EI extends into supervised prediction scenarios [18, 17, 23, 21]. Dallago et al. [8] demonstrated that straightforward fine-tuning of the representations from PLMs holds the potential to predict fitness. Hie et al. [25] used PLMs to predict the evolutionary trajectory of protein families. Yang et al. [65] advocated for using machine learning tools for small datasets and neural networks for larger datasets. In response to sparse labels, Elnaggar et al. [12] proposed a lightweight parametric model called ConvBERT to avoid overfitting. ProteinNPT [46] leveraged a non-parametric Transformer that combines masked language modeling and fitness prediction tasks, demonstrating excellence in low-resource scenarios. However, these methods often lose the generalization capabilities of PLMs after dataset-specific fine-tuning. In

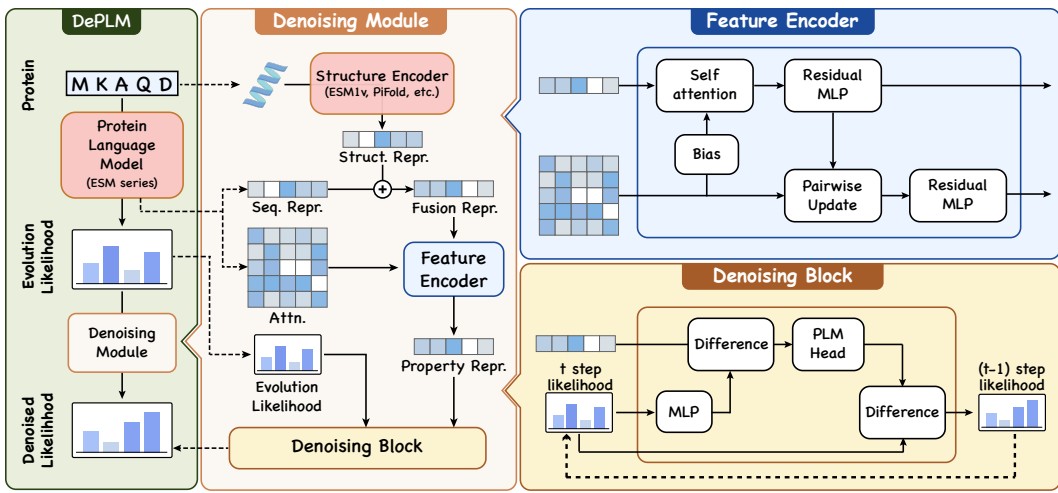

Figure 2: The architecture overview of DePLM. **Left**: DePLM utilizes evolutionary likelihoods derived from PLMs as input, and generates denoised likelihoods tailored to specific properties for predicting the effects of mutations. **Middle & Right**: Denoising module utilizes a feature encoder to derive representations of proteins, taking into account both primary and tertiary structures. These representations are then employed to filter out noise from the likelihoods via denoising blocks.

this study, we show that combining EI with experimental data in a way that minimizes reliance on dataset-specific information is crucial for enhancing performance while maintaining generalization.

**Denoising diffusion models.** Several diffusion models have been applied to protein research. Diffusion processes in discrete token spaces have been promising for designing protein sequences [3, 1, 22]. Jing et al. [32] focused on torsion for generating conformations, while Corso et al. [7] studied the product space $\mathbb{R} \times \mathrm{SO}(3) \times \mathrm{SO}(2)^m$ for protein docking. Additionally, backbone generation requires defining $\mathrm{SE}(3)$ diffusion process [66, 61, 30]. In contrast to these works, we operate the diffusion process in the rank space of mutation likelihoods to predict the fitness landscape.

## 3 Method

### 3.1 Overall Framework

DePLM, as depicted in Figure 2, is designed to filter out irrelevant information from the noisy evolutionary likelihoods produced by PLMs. Given a wildtype protein $\boldsymbol{x}^{\mathrm{wt}}$, the evolutionary likelihood produced by a PLM can be denoted as $\tilde{\boldsymbol{\Pi}} = [\tilde{\boldsymbol{\pi}}_1, \cdots, \tilde{\boldsymbol{\pi}}_n, \cdots, \tilde{\boldsymbol{\pi}}_N]$, where $\tilde{\boldsymbol{\pi}}_n \in \mathbb{R}^{20}$ denotes the probability of 20 amino acids occurring at the position $n$. This likelihood can be decomposed into the target property likelihood $\boldsymbol{\Pi}^\star$ and additive noise $\boldsymbol{\Pi}^\epsilon$ introduced by irrelevant properties, such that $\tilde{\boldsymbol{\Pi}} = \boldsymbol{\Pi}^\star + \boldsymbol{\Pi}^\epsilon$. The DePLM takes the noisy likelihood $\tilde{\boldsymbol{\Pi}}$ as input and refines it to isolate the desired likelihood $\boldsymbol{\Pi}^\star$ via a rank-based denoising diffusion process. A comprehensive explanation of the symbols and operations utilized is available in Appendix A.

### 3.2 Rank-based Denosing Diffusion Process

Denoising diffusion models consist of two main processes: a forward corruption process and a learned reverse denoising process. In the forward corruption process, small amounts of noise are progressively added to the ground truth. The reverse denoising process then learns to recover the ground truth by gradually eliminating the accumulated noise. When applying these models to denoise the mutation likelihood $\tilde{\boldsymbol{\Pi}}$ in protein optimization, however, there are two significant challenges. First, the relationship between actual property values and experimental measurements frequently exhibits nonlinearity, stemming from the diversity of experimental approaches [44]. Consequently, reliance on minimizing discrepancies between predicted and observed values for denoising purposes risks overfitting to the specific dataset utilized, thereby diminishing the model's generalization capabilities. Second, unlike those conventional denoising diffusion models, our final noisy state $\tilde{\boldsymbol{\Pi}}$ is deterministic, requiring the accumulated noise to converge [36].

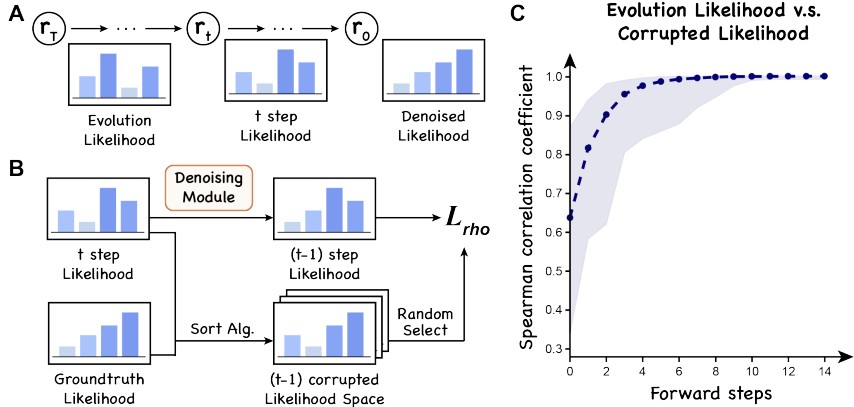

Figure 3: The training process of DePLM. **Left**: The training of DePLM involves two main steps: rank-based controlled forward corruption and learned denoising backward processes. In the corruption step, we use sorting algorithms to generate trajectories, shifting from the rank of property-specific likelihood to that of the evolutionary likelihood. DePLM is trained to model the backward process. **Right**: We illustrate the alteration of the Spearman coefficient during the transformation from evolutionary likelihood to property-specific likelihood via the sorting algorithm.

To navigate these challenges, we propose a rank-based denoising diffusion process that focuses on maximizing rank correlation (see Figure 3). Let $\boldsymbol{r}_0 = R(\boldsymbol{\Pi}^\star) \in \mathbb{N}_+^{20 \times N}$ be the rank of ground-truth likelihood of the target property, $\boldsymbol{r}_T = R(\tilde{\boldsymbol{\Pi}}) \in \mathbb{N}_+^{20 \times N}$ be the rank of noising evolutionary likelihood. Intermediate rank variables, $\boldsymbol{r}_t \in \mathbb{N}+^{20 \times N}$ for $t = 1, \ldots, T-1$, are generated along this sequence.

**Forward process.** The forward process operates in the rank space, gradually transitioning from the initial rank $\boldsymbol{r}_0$ through increasingly chaotic states $\boldsymbol{r}_t$, culminating in the final state $\boldsymbol{r}_T$ after $T$ steps. Unlike traditional models where corruption can be random, the corruption process here must carefully manage the progression because the initial state $\boldsymbol{r}_0$ and the final state $\boldsymbol{r}_T$ are pre-defined.

To manage this transition, we leverage the QuickSort algorithm [26] to create a feasible space for sampling intermediate rank variables $\boldsymbol{r}_t$. At each time step $t$, we apply the sort algorithm to generate sorting trajectories from the variable $\boldsymbol{r}_{t-1}$ towards the end variable $\boldsymbol{r}_T$. The rank variable $\boldsymbol{r}_t$ chosen along these trajectories ensures a progressive decrease in rank correlation from the initial to the intermediate states, and converse increase towards the final state. This approach ensures a controlled and meaningful progression through the rank space, as detailed in Appendix B.1. The forward process can be conceptualized as a Markov chain:

$$q(\boldsymbol{r}_{1:T-1}|\boldsymbol{r}_0, \boldsymbol{r}_T) = \prod_{t=1}^{T-1} q(\boldsymbol{r}_t|\boldsymbol{r}_{t-1}, \boldsymbol{r}_T). \tag{2}$$

**Backward process.** Given that the rank variables are non-differential, the backward process operates in the likelihood space rather than the rank space. This process constructs the likelihood $\boldsymbol{\Pi}_0$ from the noisy evolutionary likelihood $\boldsymbol{\Pi}_T = \tilde{\boldsymbol{\Pi}}$, ensuring that $R(\boldsymbol{\Pi}_0)$ equals $R(\boldsymbol{\Pi}^\star)$. Importantly, while $\boldsymbol{\Pi}_0$ and $\boldsymbol{\Pi}^\star$ are aligned in rank, they do not need to be identical in value. The key focus is on the rank variables, as the relative rankings are more critical for optimizing protein performance than the exact numerical values of the likelihoods. To facilitate this process, we employ the protein $\boldsymbol{x}^{\text{wt}}$ as a guiding signal, and model it as a conditional Markov chain with learned transitions:

$$p_\theta(R(\boldsymbol{\Pi}_{0:T-1})|\boldsymbol{x}^{\text{wt}}, \boldsymbol{\Pi}_T) \simeq p_\theta(\boldsymbol{\Pi}_{0:T-1}|\boldsymbol{x}^{\text{wt}}, \boldsymbol{\Pi}_T) = \prod_{t=1}^{T} p_\theta(\boldsymbol{\Pi}_{t-1}|\boldsymbol{x}^{\text{wt}}, \boldsymbol{\Pi}_t), \tag{3}$$

where $p_\theta(\boldsymbol{\Pi}_{t-1}|\boldsymbol{x}^{\text{wt}}, \boldsymbol{\Pi}_t)$ is the learnable transition kernel, $\theta$ denotes its parameters, and $\boldsymbol{\Pi}_t$ is the intermediate likelihood variables during the backward process, with $r_t = R(\boldsymbol{\Pi}_t)$.

**Learning objective function** The forward process introduces noise in rank space, while the reverse process denoises in likelihood space. To effectively link these two processes, we leverage Spearman's

rank correlation $\rho$ defined in Eq. (1). Building upon this, we modify the variational Evidence Lower BOund (ELBO) as our learning objective function:

$$\mathbb{E}[\log p_\theta(\boldsymbol{r}_0|\boldsymbol{x}^{\text{wt}})] = \mathbb{E}\left[\log \mathbb{E}_{q(\boldsymbol{r}_{1:T-1}|R(\boldsymbol{\Pi}^\star),R(\tilde{\boldsymbol{\Pi}}))}\frac{p_\theta(\boldsymbol{r}_{0:T}|\boldsymbol{x}^{\text{wt}})}{q(\boldsymbol{r}_{1:T-1}|R(\boldsymbol{\Pi}^\star),R(\tilde{\boldsymbol{\Pi}}))}\right]$$

$$\geq -\mathbb{E}_q\left[\sum_{t=1}^{T}\left(1 - \rho\left(q(\boldsymbol{r}_{t-1}|R(\boldsymbol{\Pi}_0),R(\boldsymbol{\Pi}_t))||p_\theta(R(\boldsymbol{\Pi}_{t-1})|\boldsymbol{x}^{\text{wt}},\boldsymbol{\Pi}_t))\right)\right]. \quad (4)$$

Detailed derivations are provided in the Appendix B.2. Overall, given a protein $\boldsymbol{x}^{\text{wt}}$, its property likelihood $\boldsymbol{\Pi}^\star$ is generated by first drawing evolutionary likelihood $\bar{\boldsymbol{\Pi}}$ from PLMs, and then iteratively refined through $p_\theta(\boldsymbol{\Pi}_{t-1}|\boldsymbol{x}^{\text{wt}},\boldsymbol{\Pi}_t)$. We predict $y$ according to Eq. 10 in Appendix C.1.

### 3.3 Implementation of Denoising Markov Kernel

To parameterize the transition kernel $p_\theta(\boldsymbol{\Pi}_{t-1}|\boldsymbol{x}^{\text{wt}},\boldsymbol{\Pi}_t)$, as shown in the Denoising Module of Figure 2, DePLM first learns a protein representation (Feature Encoder of Figure 2), and then utilizes it to guide the process of discerning and eliminating noise (Denoising Block of Figure 2).

**Feature encoder**. We encode features from both sequences and structures because they complementarily describe the impact of mutations. Sequence information is derived from representations $\boldsymbol{h}_e$ and attention weights $\boldsymbol{M} \in \mathbb{R}^{N \times N \times m}$ generated by PLMs, where $m$ is the head number. Meanwhile, structural information is obtained through trained structure encoders like ESM-IF [28], which process the protein backbone to produce structural representations $\boldsymbol{h}_s$. We merge the two sets of representation using Multi-Layer Perceptrons (MLPs), yielding a unified representation $\boldsymbol{h}$ in $\mathbb{R}^{N \times d}$, where $d$ represents the hidden dimension.

The feature encoder leverages multiple stacked layers to update $\boldsymbol{h}$ and $\boldsymbol{M}$. Let's denote the output feature vectors of representations and attention weights in layer $l$ as $\boldsymbol{h}^l$ and $\boldsymbol{M}^l$, respectively. Initially, $\boldsymbol{h}^0$ equals $\boldsymbol{h}$ and $\boldsymbol{M}^0$ equals $\boldsymbol{M}$. We update the representations using the standard attention mechanism, incorporating a bias derived from the attention weights:

$$\boldsymbol{h}^{l+1} = \text{ResidualMLP}(\text{Softmax}(\frac{\boldsymbol{q}^l(\boldsymbol{k}^l)^T + \boldsymbol{B}_M^l}{\sqrt{d}})\boldsymbol{v}^l), \quad (5)$$

where $\boldsymbol{q}^l$, $\boldsymbol{k}^l$, and $\boldsymbol{v}^l$ are the linear projection of $\boldsymbol{h}^l$ and $\boldsymbol{B}_M^l$ is the linear projection of $\boldsymbol{M}^l$. Then, the attention weights are updated by communicating with the sequence representation through both an outer product $\otimes$ and an outer difference $\ominus$:

$$\boldsymbol{B}_h^{l+1} = (\boldsymbol{q}^{l+1} \otimes \boldsymbol{k}^{l+1})||(\boldsymbol{q}^{l+1} \ominus \boldsymbol{k}^{l+1}), \quad (6)$$

$$\boldsymbol{M}^{l+1} = \text{ResidualMLP}(\boldsymbol{M}^l + \boldsymbol{B}_h^{l+1}), \quad (7)$$

where $||$ means the concatenation operation. The final layer output $\boldsymbol{h}^L$ is used in the denoising block.

**Denosing block**. Given the intermediate likelihood variable at the $t$ step $\boldsymbol{\Pi}_t$, we employ a denoising block to implement Eq. (3). The central premise is that $\boldsymbol{h}^L$ should encapsulate only property-specific protein information. By subtracting $\boldsymbol{h}^L$ from the hidden representation of the noisy $\boldsymbol{\Pi}_t$, we isolate the hidden representation of the noise. This hidden representation is then transformed into the likelihood space using a PLM head (PLMHead),

$$\boldsymbol{\Pi}_{t-1} = \boldsymbol{\Pi}_t - \text{PLMHead}(\text{MLP}(\boldsymbol{\Pi}_t) - \boldsymbol{h}^L). \quad (8)$$

Here $\text{MLP}(\boldsymbol{\Pi}_t)$ denotes a trainable MLP that maps $\boldsymbol{\Pi}_t$ to the hidden representation space, while $\text{PLMHead}$ maps hidden presentations back to the likelihood space. It is important to use a frozen $\text{PLMHead}$ to ensure consistency between noise and noisy likelihoods in a unified space. This approach effectively denoises $\boldsymbol{\Pi}_t$ to obtain $\boldsymbol{\Pi}_{t-1}$.

## 4 Experiments

In this section, we extensively evaluate DePLM across various datasets and demonstrate its superior performance and robust generalization capabilities. Specifically, we aim to address the following key questions. Performance comparison (Q1): Can DePLM beat SOTA on protein fitness prediction tasks? Generalization ability (Q2): Does DePLM maintain its generalization ability post-training? Ablation study (Q3): What is the extent of improvement achievable for each component? Analysis (Q4): Does the assumption that EI contains noise hold?

Table 1: Model performance on protein engineering tasks. The **best** and suboptimal results are labeled with bold and underline, respectively. ProteinGym results of OHE, $\overline{\text{ESM-MSA}}$, Tranception, and ProteinNPT are borrowed from Notin et al. [46]. Other results are obtained by our own experiments.

| Model | ProteinGym | | | | | $\beta$-lact. | GB1 | Fluo. |
|---|---|---|---|---|---|---|---|---|
| | Stability | Fitness | Expression | Binding | Activity | | | |
| CNN | 0.788 | 0.588 | 0.627 | 0.599 | 0.573 | 0.781 | 0.502 | **0.682** |
| ResNet | 0.734 | 0.489 | 0.521 | 0.525 | 0.481 | 0.152 | 0.133 | 0.636 |
| LSTM | 0.745 | 0.413 | 0.477 | 0.496 | 0.408 | 0.139 | -0.002 | 0.494 |
| Transformer | 0.560 | 0.149 | 0.156 | 0.172 | 0.155 | 0.261 | 0.271 | 0.643 |
| OHE | 0.718 | 0.545 | 0.573 | 0.562 | 0.555 | 0.823 | 0.533 | 0.657 |
| ESM-1v | 0.880 | 0.566 | 0.642 | 0.596 | 0.572 | 0.536 | 0.394 | 0.438 |
| ESM-2 | 0.882 | 0.573 | 0.645 | 0.587 | 0.576 | - | - | - |
| ESM-MSA | 0.885 | 0.568 | 0.632 | 0.565 | 0.600 | - | - | - |
| ProtSSN | 0.877 | 0.692 | 0.718 | 0.757 | 0.678 | - | - | - |
| SaProt | 0.882 | 0.686 | 0.716 | 0.749 | 0.677 | - | - | - |
| Tranception | 0.871 | 0.632 | 0.704 | 0.671 | 0.623 | - | - | - |
| ProteinNPT | **0.904** | 0.668 | 0.736 | 0.706 | 0.680 | - | - | - |
| DePLM (ESM1v) | 0.887 | 0.704 | 0.738 | **0.773** | 0.688 | 0.900 | **0.676** | 0.662 |
| DePLM (ESM2) | 0.897 | **0.707** | **0.742** | 0.764 | **0.693** | **0.904** | 0.665 | 0.662 |

## 4.1 Experimental Setup

We begin by outlining the general experimental setups used in our evaluations. We use ESM-IF [28] as the structure encoder and the structures are predicted by AlphaFold2 [33]. DePLM comprises 42.2 million trainable parameters and involves 3 diffusion steps. We set the learning rate at 0.0001, with a weight decay of 0.005, utilizing AdamW as the optimizer. All models are trained on four Nvidia V100 32G GPUs for up to 100 epochs by default.

## 4.2 Performance Comparison (Q1)

**Datasets and Baselines.** We conducted a thorough study across four benchmarks, including ProteinGym [44], $\beta$-Lactamase (Abbr., $\beta$-lact.) and Fluorescence (Abbr., Fluo.) from PEER [63], and GB1 (utilizing a *2-vs-rest* split) from FLIP [8], where the latter two involve multiple mutants. We compare DePLM with nine baselines, including 1) four protein sequence encoders trained from scratch (CNN [53], ResNet [47], LSTM [47], and Transformer [47]) as naive baselines, 2) five extended baselines that incorporate self-supervised models (OHE [27], fine-tuned versions of ESM-1v [39], ESM-MSA [48], and Tranception [43], as well as ProteinNPT [46]). More details about datasets and baselines can be found in Appendix C.1 and C.2.

**Results.** We present the evaluation results in Table 9. DePLM achieves better performance compared to the baselines, affirming the advantage of integrating evolutionary information with experimental data for protein engineering tasks. It is worth noting that ESM-MSA and Tranception exhibit enhanced EI compared to ESM-1v due to the introduction of MSAs. By comparing their results, we demonstrate that higher-quality EI significantly improves outcomes after fine-tuning. However, even with these improvements, their performance still falls short of that achieved by DePLM. We attribute this difference to the architecture employed by our model, which enables more efficient utilization of experimental data. We also notice that DePLM yields better performance than ProteinNPT, underscoring the efficacy of the proposed denoising training process.

## 4.3 Generalization Ability (Q2)

**Datasets and Baselines.** Computational techniques that can generalize across different proteins are essential, given the limitations of DMS experimental methods in handling various proteins [58]. In our study, we leverage ProteinGym to test this generalization capability. Specifically, ProteinGym categorizes DMS datasets into five coarse categories based on the protein properties they measure: stability, fitness, expression, binding, and activity. Given a testing dataset, we randomly select an additional 40 datasets from the same category for training. Importantly, we ensure that the sequence similarity between the datasets used for training and testing is kept below 50% to prevent potential data leakage. In this experiment, we compare DePLM with four self-supervised baselines (ESM-1v [39], ESM-2 [34], and TranceptEVE [45]), two structure-based baselines (ESM-IF [28]

Table 2: Generalization ability evaluation. The **best** and suboptimal results are labeled with bold and underline, respectively. The information (evolutionary, structural or experimental) involved in each model is provided. Results of unsupervised methods are borrowed from Notin et al.[43]. Other results are obtained by our own experiments. (FT=Fine-tuned version)

| Model | Information | | | ProteinGym | | | | |
|---|---|---|---|---|---|---|---|---|
| | Evo. | Struct. | Exp. | Stability | Fitness | Expression | Binding | Activity |
| ESM1v | ✓ | | | 0.437 | 0.395 | 0.427 | 0.287 | 0.415 |
| ESM2 | ✓ | | | 0.523 | 0.396 | 0.439 | 0.356 | 0.433 |
| ProtSSN | ✓ | ✓ | | 0.560 | 0.408 | 0.435 | 0.362 | 0.458 |
| TranceptEVE L | ✓ | | | 0.500 | 0.477 | 0.457 | 0.360 | 0.487 |
| ESM-IF | | ✓ | | 0.624 | 0.346 | 0.436 | 0.380 | 0.412 |
| ProteinMPNN | | ✓ | | 0.564 | 0.166 | 0.209 | 0.159 | 0.203 |
| CNN | | | ✓ | 0.141 | 0.053 | 0.043 | 0.056 | 0.095 |
| ESM1v (FT) | ✓ | | ✓ | 0.497 | 0.318 | 0.301 | 0.216 | 0.385 |
| ESM2 (FT) | ✓ | | ✓ | 0.454 | 0.359 | 0.338 | 0.276 | 0.391 |
| ProtSSN (FT.) | ✓ | ✓ | ✓ | 0.689 | 0.448 | 0.478 | 0.421 | 0.488 |
| SaProt (FT.) | ✓ | ✓ | ✓ | 0.703 | 0.442 | 0.496 | 0.391 | 0.495 |
| DePLM (ESM1v) | ✓ | ✓ | ✓ | 0.763 | 0.467 | 0.506 | 0.409 | 0.499 |
| DePLM (ESM2) | ✓ | ✓ | ✓ | **0.773** | **0.480** | **0.510** | **0.441** | **0.518** |

Table 3: Ablation study of the modules in DePLM.

| Method | ProteinGym (Valid) | GB1 | Fluorescence | Average |
|---|---|---|---|---|
| DePLM | **0.690** | 0.665 | **0.662** | **0.672** |
| w/o structural information | 0.683 | **0.672** | 0.659 | 0.671 |
| w/o feature encoder | 0.682 | 0.659 | 0.661 | 0.667 |
| w/o denoising block | 0.656 | 0.644 | 0.655 | 0.652 |
| w/o rank objective | 0.322 | 0.588 | 0.552 | 0.487 |

and ProteinMPNN [9]), and three supervised baselines (CNN [8], fine-tuned version of ESM-1v and ESM-2). The choice of baselines is elaborated in Appendix C.1.

**Results.** As shown in Table 2, one can observe that DePLM consistently outperforms all baseline models. This finding underscores the inadequacy of baselines that rely solely on unfiltered evolutionary information, which often dilutes target properties due to concurrent optimization of multiple objectives. By eliminating the influence of irrelevant factors, DePLM enhances its performance significantly. In addition, the performance of baselines trained to minimize the disparity between predicted and experimental scores (ESM1v (FT) and ESM2 (FT)) falls significantly short of that achieved by our DePLM. This observation highlights that optimizing the model in a rank space introduces less bias from specific datasets and yields superior generalization. Furthermore, we observe that protein structure information contributes to the stability and binding properties, whereas evolutionary information enhances fitness and activity attributes.

## 4.4 Ablation Study (Q3)

The ablation study aims to validate the efficacy of the modules devised for DePLM. These modules include the use of structural information, the feature encoder, the denoising module, and the rank objective. The results in Table 8 show the absence of any of these modules leads to a decline in performance, indicating their collective significance. Additionally, we observe that the rank objective has the largest impact on performance, highlighting the importance of reducing dataset-specific information. Further ablation studies are detailed in Appendix C.4.

## 4.5 Analysis and Discussion (Q4)

**Necessity of Filtering Property-Irrelevant Information in EI.** To ascertain the importance of filtering out property-irrelevant information, we analyzed the impact of training with datasets targeting various optimization objectives. As illustrated in Figure 4 (left), we observe that using training datasets with characteristics divergent from those of the test dataset leads to diminished performance across stability, expression, and activity properties. This reduction in performance

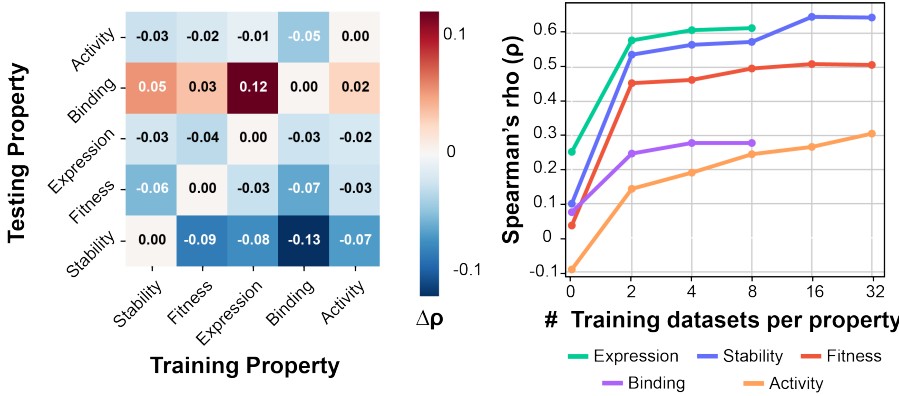

Figure 4: Visualization of the impact of optimization targets and size of training data on performance.

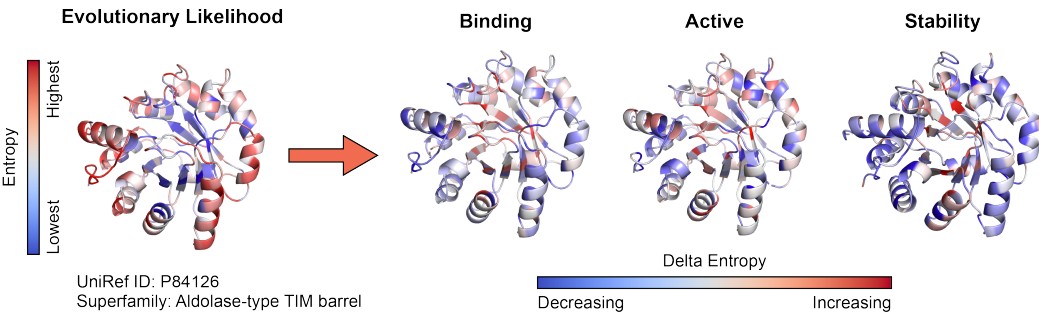

Figure 5: Visualization of the impact of denoising process on the evolutionary likelihood.

underscores the detrimental interference among different properties, emphasizing the need to mitigate these adverse effects. Interestingly, we observed that training on datasets targeting alternative properties can improve performance in the binding property. This improvement is likely due to the scarce availability of datasets specifically focusing on the binding property within ProteinGym. This observation suggests a beneficial cross-utilization of data, where leveraging information from unrelated attributes can enhance inference for properties with limited data. In Figure 4 (right), we illustrate the influence of dataset size on performance. We found that even with a minimal number of datasets (K=2), DePLM significantly boosts performance, indicating its proficiency in filtering out irrelevant information. Furthermore, as the number of training datasets increases, there is a corresponding improvement in performance, showcasing the model's capability to continuously enhance its filtering efficacy with more data.

**Differentiating the Noisy Evolutionary Likelihood from the Property-specific Likelihood.** Denoising the evolutionary likelihood is helpful in identifying protein sequences that manifest specific properties. In Figure 5 (left), we utilize entropy to gauge the degree of conservation at each position and illustrate the likelihood attributed to hydrophobic amino acids within the structure. Residues with outward-facing side chains on alpha helices are associated with higher entropy, whereas inward-facing positions exhibit lower entropy. Figure 5 (right) compares the evolutionary likelihood and the property-specific likelihood and visualizes the differences. We notice that binding and active share similar offsets. This suggests that the indole-3-glycerolphosphate synthase functions by attaching to other molecules, which makes sense considering its role as an enzyme [54]. We also observe that the property-specific likelihoods exhibit a more uniform entropy when compared to the noisy evolutionary likelihood. We assume this may arise due to a bias towards inward-facing positions across all properties, potentially overshadowing the evolutionary significance of outward-facing side chains. This imbalance is addressed through the denoising process described in Section 3.2.

# 5 Conclusion and Limitations

In this paper, we propose DePLM, a simple yet effective fine-tuning approach that leverages a feature encoder to obtain expressive protein representations and then uses them to extract property-specific likelihood from the noisy evolutionary likelihood for mutational effect prediction. Our experiments demonstrate that DePLM not only surpasses state-of-the-art baselines but also shows exceptional generalization capabilities. Additionally, our analysis confirms that utilizing sufficiently large datasets or incorporating data from other relevant properties can significantly enhance performance.

Due to limited resources, our experiments are conducted using wild-type marginal probability. This approach predicts the impacts of all mutations in a single forward pass, summing the effects of individual mutations to estimate the consequences of multiple mutations simultaneously. However, this method is not ideal, as it overlooks the complex interactions between mutations. Our method can potentially achieve better performance in predicting the effects of multiple mutations, leveraging more effective prediction techniques such as masked marginal probability.

## Acknowledgments and Disclosure of Funding

We would like to thank all anonymous reviewers for their insightful and invaluable comments. This work is supported by National Natural Science Foundation of China (U23A20496, 62302433, 62301480), Zhejiang Provincial "Jianbing" "Lingyan" Research and Development Program of China (2024C01135), Hangzhou West Lake Pearl Project Leading Innovative Youth Team Project (TD2023017), Zhejiang Provincial Natural Science Foundation of China (LQ24F020007) and CCF-Tencent Rhino-Bird Fund (RAGR20230122).

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

# A  Definitions of symbols

To ease reading and facilitate understanding of our DePLM, in Table 4, we summarize the symbols and notations employed throughout the paper.

Table 4: Definitions of symbols used in this paper.

| Symbol | Description |
|---|---|
| $l, m, n, t$ | the index number |
| $\boldsymbol{x}^{\text{wt}}$ | the sequence of a wildtype protein |
| $N$ | the number of amino acids in a protein sequence |
| $d$ | the hidden dimension of DePLM |
| $\boldsymbol{h}_e, \boldsymbol{h}_s, \boldsymbol{h}$ | the representations of wildtype protein sequence, structure and fusion of the two |
| $\boldsymbol{M}$ | the attention weights produced by PLMs using the wildtype protein sequence as input |
| $\boldsymbol{h}^l, \boldsymbol{M}^l$ | the representations and attention weights produced by layer $l$ |
| $\tilde{\boldsymbol{\Pi}}$ | the evolutionary likelihood produced by PLMs using the wildtype protein sequence as input |
| $\boldsymbol{\Pi}^\star$ | the likelihood corresponding to the property of interest |
| $\boldsymbol{\Pi}^\epsilon$ | the noise likelihood influenced by irrelevant properties |
| $\boldsymbol{\Pi}_{0:T}$ | the intermediate likelihood variables during the backward process |
| $R(\cdot)$ | the rank function for fitness scores |
| $T$ | the number of diffusion steps |
| $\boldsymbol{r}_{0:T}$ | the rank variables during the forward process |
| $q(\boldsymbol{r}_{t-1}|\boldsymbol{r}_0, \boldsymbol{r}_t)$ | the forward process |
| $p_\theta(\boldsymbol{\Pi}_{t-1}|\boldsymbol{x}^{\text{wt}}, \boldsymbol{\Pi}_t)$ | the learnable transition kernel parameterized by $\theta$ during the reverse process |
| $\mathbb{E}$ | the probability that denoising reaches the ground truth |
| $\mathbb{R}$ | the real number space |
| $\mathbb{N}_+$ | the positive integer space |
| $\rho$ | the Spearman's rank correlation coefficient |

# B  Method details

## B.1  Constructing the Space of Rank Variables

We provide the pseudo codes as follows, to help readers easily understand how to construct the space of rank variables in the forward process.

---

**Algorithm 1** Constructing the Space of Rank Variables

---

**Data:** The ranks of likelihoods at time steps $t_1$ and $t_2$ (where $t_1 < t_2$), represented by $\boldsymbol{r}_{t_1}$ and $\boldsymbol{r}_{t_2}$; the number of sampling trajectories $\eta$.
**Result:** The feasible space of rank variables $\mathbb{S}^{\boldsymbol{r}}_{t_1:t_2}$ between $\boldsymbol{r}_{t_1}$ and $\boldsymbol{r}_{t_2}$.

$\mathbb{S}^{\boldsymbol{r}}_{t_1:t_2} \leftarrow \varnothing, \xi \leftarrow \varnothing, i \leftarrow 0.$ // Variable initialization
Compute sorting index $\boldsymbol{I}_{t_1}$ so that $\boldsymbol{r}_{t_1}[\boldsymbol{I}_{t_1}]$ is monotonically increasing and $\boldsymbol{r}_{t_1}[\boldsymbol{I}_{t_1}][\boldsymbol{I}_{t_1}^{-1}] = \boldsymbol{r}_{t_1}$.
$\boldsymbol{r}_{t_1} \leftarrow \boldsymbol{r}_{t_1}[\boldsymbol{I}_{t_1}], \boldsymbol{r}_{t_2} \leftarrow \boldsymbol{r}_{t_2}[\boldsymbol{I}_{t_1}].$
$\xi \leftarrow \xi \cup \{[0, \text{len}(\boldsymbol{r}_{t_1}) - 1]\}$ // Set left index $\phi$ to 0 and right index $\psi$ to $\text{len}(\boldsymbol{r}_{t_1}) - 1$
**while** $i < \eta$ **do**
  **while** $\text{Stack} \neq \varnothing$ **do**
    $\tau \leftarrow \varnothing.$
    **for** $[\phi, \psi] \in \xi$ **do**
      $\boldsymbol{r}_{t_2}, \varphi = \text{Sort}(\boldsymbol{r}_{t_2}, \phi, \psi)$ // No element in $[\phi, \varphi]$ is greater than any element in $[\varphi, \psi]$.
      $\tau \leftarrow \tau \cup \{[\phi, \varphi - 1]\}$ if $\phi < \varphi - 1.$
      $\tau \leftarrow \tau \cup \{[\varphi + 1, \psi]\}$ if $\psi > \varphi + 1.$
    **end for**
    $\xi \leftarrow \tau$
    $\mathbb{S}^{\boldsymbol{r}}_{t_1:t_2} \leftarrow \mathbb{S}^{\boldsymbol{r}}_{t_1:t_2} \cup \boldsymbol{r}_{t_2}[\boldsymbol{I}_{t_1}^{-1}]$
  **end while**
  $i \leftarrow i + 1$
**end while**

---

## B.2  Derivation of variational lower bound (ELBO) details

The ELBO can be calculated as follows:

$$
\begin{aligned}
\mathbb{E}[\log p_\theta(\boldsymbol{r}_0|\boldsymbol{x}^{\mathrm{wt}})] &= \mathbb{E}\left[\log \mathbb{E}_{q(\boldsymbol{r}_{1:T-1}|R(\boldsymbol{\Pi}^\star),R(\tilde{\boldsymbol{\Pi}}))} \frac{p_\theta(\boldsymbol{r}_{0:T}|\boldsymbol{x}^{\mathrm{wt}},\tilde{\boldsymbol{\Pi}}) \times p(\tilde{\boldsymbol{\Pi}})}{q(\boldsymbol{r}_{1:T-1}|R(\boldsymbol{\Pi}^\star),R(\tilde{\boldsymbol{\Pi}}))}\right] \\
&\geq \mathbb{E}_q \log \frac{p_\theta(\boldsymbol{r}_{0:T}|\boldsymbol{x}^{\mathrm{wt}},\boldsymbol{\Pi}_T) \times p(\boldsymbol{\Pi}_T)}{q(\boldsymbol{r}_{1:T-1}|R(\boldsymbol{\Pi}_0),R(\boldsymbol{\Pi}_T))} \\
&= \mathbb{E}_q\left[\log p(\boldsymbol{\Pi}_T) - \sum_{t=1}^{T} \log \frac{p_\theta(R(\boldsymbol{\Pi}_{t-1})|\boldsymbol{x}^{\mathrm{wt}},\boldsymbol{\Pi}_t)}{q(\boldsymbol{r}_t|R(\boldsymbol{\Pi}_{t-1}),R(\boldsymbol{\Pi}_T))}\right] \\
&= \mathbb{E}_q\left[\log p(\boldsymbol{\Pi}_T) - \log \frac{p_\theta(R(\boldsymbol{\Pi}_0)|\boldsymbol{x}^{\mathrm{wt}},\boldsymbol{\Pi}_1)}{q(\boldsymbol{r}_1|R(\boldsymbol{\Pi}_0),R(\boldsymbol{\Pi}_T))} \right. \\
&\qquad \left. - \sum_{t=2}^{T}\left(\log \frac{p_\theta(R(\boldsymbol{\Pi}_{t-1})|\boldsymbol{x}^{\mathrm{wt}},\boldsymbol{\Pi}_t)}{q(\boldsymbol{r}_{t-1}|R(\boldsymbol{\Pi}_0),R(\boldsymbol{\Pi}_t))} + \log \frac{q(\boldsymbol{r}_{t-1}|R(\boldsymbol{\Pi}_0),R(\boldsymbol{\Pi}_T))}{q(\boldsymbol{r}_t|R(\boldsymbol{\Pi}_0),R(\boldsymbol{\Pi}_T))}\right)\right] \\
&= \mathbb{E}_q\left[\log \frac{p(R(\boldsymbol{\Pi}_T))}{q(\boldsymbol{r}_T|R(\boldsymbol{\Pi}_0),R(\boldsymbol{\Pi}_T))} - \log p_\theta(R(\boldsymbol{\Pi}_0)|\boldsymbol{x}^{\mathrm{wt}},\boldsymbol{\Pi}_1) \right. \\
&\qquad \left. - \sum_{t=2}^{T} \log \frac{p_\theta(R(\boldsymbol{\Pi}_{t-1})|\boldsymbol{x}^{\mathrm{wt}},\boldsymbol{\Pi}_t)}{q(\boldsymbol{r}_{t-1}|R(\boldsymbol{\Pi}_0),R(\boldsymbol{\Pi}_t))}\right] \\
&= -\mathbb{E}_q\left[(1-\rho(q(\boldsymbol{r}_T|R(\boldsymbol{\Pi}_0),R(\boldsymbol{\Pi}_T))||p(R(\boldsymbol{\Pi}_T))) \right. \\
&\qquad - \log p_\theta(R(\boldsymbol{\Pi}_0)|\boldsymbol{x}^{\mathrm{wt}},\boldsymbol{\Pi}_1)) \\
&\qquad \left. + \sum_{t=2}^{T}\left(1-\rho(q(\boldsymbol{r}_{t-1}|R(\boldsymbol{\Pi}_0),R(\boldsymbol{\Pi}_t))||p_\theta(R(\boldsymbol{\Pi}_{t-1})|\boldsymbol{x}^{\mathrm{wt}},\boldsymbol{\Pi}_t)))\right]
\end{aligned}
$$

The first term, $1 - \rho(q(\boldsymbol{r}_T|R(\boldsymbol{\Pi}_0),R(\boldsymbol{\Pi}_T))||p(R(\boldsymbol{\Pi}_T))$, is a constant and can therefore be omitted from the objective function. Following Xu et al. [64], we combine the second term, $\log p_\theta(R(\boldsymbol{\Pi}_0)|\boldsymbol{x}^{\mathrm{wt}},\boldsymbol{\Pi}_1))$, with the final term. This leads us to the expression for ELBO $\mathcal{L}_{\mathrm{ELBO}} = \sum_{t=1}^{T}(1-\rho(q(\boldsymbol{r}_{t-1}|R(\boldsymbol{\Pi}_0),R(\boldsymbol{\Pi}_t))||p_\theta(R(\boldsymbol{\Pi}_{t-1})|\boldsymbol{x}^{\mathrm{wt}},\boldsymbol{\Pi}_t)))$ as in Eq. 4.

## C  Experimental details

### C.1  Baseline details

#### C.1.1  Self-supervised methods

Self-supervised methods can decode the link between protein sequences and their function by analyzing evolutionary-scale data. This is because protein properties influence the choice of sequences throughout evolution. Leveraging advancements from Natural Language Processing (NLP), researchers have leveraged the Transformer architectures combined with the Masked Language Modeling (MLM) learning objective to effectively capture the complexity of protein sequences. During training, each input sequence $\boldsymbol{x}$ is altered by replacing some amino acids with a special mask token. The network learns to predict these missing tokens from the modified sequence:

$$
\mathcal{L}_{\mathrm{MLM}} = \mathbb{E}_{\boldsymbol{x}\sim X}\left[\mathbb{E}_M\left[\sum_{m\in M} -\log p(\boldsymbol{x}_m|\boldsymbol{x}_{/M})\right]\right], \tag{9}
$$

where $M$ represents a set of masked positions and $\boldsymbol{x}_{/M}$ denotes the masked protein sequence. Specifically, MLM replaces 80% of these positions with the special "[MASK]" token, 10% with a randomly chosen alternative amino acid token, and 10% remain as the original input tokens.

With the MLM learning objective, PLMs can output the likelihood of each amino acid occurring at masked positions given the context of unmasked amino acids. This can be directly applied to the

prediction of mutational effects by comparing the likelihood of the mutant sequence to the wildtype sequence [39]. Specifically, one uses the amino acid in the wildtype protein as a reference state:

$$y \simeq \sum_{m \in M} \log p(\boldsymbol{x}_m = \boldsymbol{x}_m^{\text{mt}} | \boldsymbol{x}_{/M}^{\text{wt}}) - \log p(\boldsymbol{x}_m = \boldsymbol{x}_m^{\text{wt}} | \boldsymbol{x}_{/M}^{\text{wt}}). \tag{10}$$

The probability $p$ in Eq. 10 is actually implemented as $\boldsymbol{\Pi}^*$ with our DePLM framework. Note that although the property $y$ (e.g., thermostability) has its own physical concepts and units, which are usually different from the likelihood on the right-hand side of Eq. 10, one usually cares the relative ranks of property values. Hence the above equivalence holds.

### C.1.2 Supervised methods

Supervised methods use experimental data to fit neural networks for predicting protein fitness landscapes. Most existing methods treat fitness landscape prediction as a regression problem. The process begins by transforming the wildtype sequence $\boldsymbol{x}^{\text{wt}}$ and its mutations $(\mu_1, \ldots, \mu_n)$ into mutant sequences $(\boldsymbol{x}_1^{\text{mt}}, \ldots, \boldsymbol{x}_n^{\text{mt}})$. These sequences are then input into a protein encoder to extract features. A predictor is subsequently employed to estimate the property value $y$, optimized using mean squared error (MSE) loss.

### C.1.3 Baselines

**Performance Comparison.** We adopt nine supervised baselines to compare with DePLM. Following Xu et al. [63], we use four well-known protein sequences encoders: shallow CNN and ResNet, which focus on short-range interactions, and LSTM and Transformer, which focus on long-range interactions. Subsequently, we utilize ridge regression to forecast fitness using features extracted by the aforementioned models. We summarize each model in Table 5

Table 5: Baseline model descriptions. *Abbr.*, Params.: parameters; dim.: dimension.

| Model | Input Layer | Hidden Layers | #Params. |
|---|---|---|---|
| shallow CNN | 21-dim. one-hot residue type | 1× 1D conv. layers (hidden dim.: 1024; kernel size 7; stride: 1; padding: 3) | 2.7M |
| ResNet | 21-dim. one-hot residue type | 8 × residual blocks (hidden dim.: 512; kernel size 3; stride: 1; padding: 1) | 11.0M |
| LSTM | 21-dim. one-hot residue type | 3× bidirectional LSTM layers (hidden dim.: 640) | 26.7M |
| Transformer | 21-dim. one-hot residue type | 3× Transformer blocks (hidden dim.: 512; #attn heads: 8; activation: GELU | 21.3M |

We also compare DePLM with five extended baselines that incorporate self-supervised models. Specifically, we investigate OHE [27], which utilizes ridge regression supplemented with protein likelihood from DeepSequence [49]. We also assess fine-tuned versions of ESM-1v [39], ESM-MSA [48], and Tranception [43], which predicts the fitness score by inputting their representations and likelihoods as features into an trainable MLPs. Furthermore, our approach is compared against ProteinNPT [46], a state-of-the-art baseline equipped with a non-parametric transformer specifically designed for label-scarce settings. To ensure fair comparisons, we keep the parameters of these unsupervised models frozen.

**Generalization ability**. We conduct a comprehensive comparison of model generalization capabilities, including three types of models. 1) Self-supervised models trained on sequence variations. Among these, ESM-1v [39] and ESM-2 [34] are leading PLMs based on MLM objective, which form the foundation of DePLM. ProtSSN [56] integrates both evolutionary and structural features, enhancing the model's ability to capture complex protein characteristics. We also assess TranceptEVE [45], the current state-of-the-art in unsupervised ProteinGym benchmarking. 2) Inverse Folding models. Since the function on proteins is dependent on its shape, amino acid likelihoods derived based on backbone can be used to infer mutation effects. Therefore, we employ ESM-IF [28] and ProteinMPNN [9] as baseline models. 3) Supervised models. We also emply three supervised baselines, including CNN, which has previously been acknowledged as the best naive baseline. Due to computational

resource limitations, we excluded MSA-based methods (ESM-MSA [48], Tranception [43], and ProteinNPT [46]), leaving the fine-tuned version ESM-1v and ESM-2.

## C.2 Dataset details

ProteinGym is an extensive set of Deep Mutational Scaning (DMS) assays, containing 217 datasets. Due to the length limitations of PLMs, we excluded datasets involving wildtype proteins longer than 1024, leaving us with 201 DMS datasets. ProteinGym categorizes DMS into five coarse categories: 66 for stability, 69 for fitness, 16 for expression, 12 for binding, and 38 for activity.

**Performance Comparison**. We implemented the *Random* cross-validation method recommended by [46]. In this approach, each mutation in the dataset is randomly assigned to one of five folds. The model's performance is then evaluated by averaging the results across these five folds.

**Generalization ability**. Given a testing dataset, we randomly select up to 40 datasets consistent with its optimization target (e.g., thermostability) as training data. We ensure that the sequence similarity between the training protein and the test protein is less that 50% to avoid data leakage.

**Ablation**. Following Notin et al. [46], we select 8 datasets for ablation study:

- BLAT_ECOLX_Jacquier_2013 [31]
- CALM1_HUMAN_Weile_2017 [62]
- DLG4_RAT_McLaughlin_2012 [38]
- DYR_ECOLI_Thompson_2019 [57]
- P53_HUMAN_Giacomelli_2018 [19]
- REV_HV1H2_Fernandes_2016 [13]
- RL40A_YEAST_Roscoe_2013 [52]
- TAT_HV1BR_Fernandes_2016 [13]

Detailed DMS-level performance on random cross-validation scheme and generalization setting is reported in Figure 7 and Figure 8. The error bars for the random cross-validation scheme are also reported in Table 9.

Table 6: Comparison of the computational costs of DePLM and ProteinNPT, using A4GRB6_PSEAI as an example. This protein has 267 residues and 5001 mutants. (All results were obtained using the ptflops package.) The training computational cost = (one forward computational cost) × (number of mutants / number of predicted mutants per forward) × (number of epochs.). The total number of parameters = non-trainable parameters + trainable parameters. For DePLM, 180.56 GMACs is the computational cost of the sequence encoder, and 77.55 GMACs is the computational cost for extracting structural information.

| Method | Training (MACs) | Inference (MACs) | Parameter (#) |
|---|---|---|---|
| ProteinNPT | 5.85B = 11724.82 × 5001 × 100 | 58.5M = 11724.82 × 5001 | 100M + 119M |
| DePLM | **9.16K** = 180.56 + 77.55 + 89.05 × 100 | **347.16** = 180.56 + 77.55 + 89.05 | 792M + 42.2M |

## C.3 License details

In this paper, we utilized several advanced protein language models: ESM-1v [39] [MIT License], ESM-2 [34] [MIT License], and ProteinNPT [46] [MIT License]. For our datasets, we employed ProteinGym [44] [MIT License], PEER [63] [Apache-2.0 license], FLIP [8] [AFL-3.0 license].

## C.4 Supplementary Ablation Study

**Computational cost** DePLM utilizes the QuickSort algorithm, which has a time complexity of $O(n \log n)$. Given the sparsity of labels, sorting the assay with the most labels in ProteinGym ( 500k mutants) takes only 1.45 seconds on 2.8GHz Quad-Core Intel Core i7, while sorting the assay with the median number of labels ( 5k mutants) takes just 0.0056 seconds.

By leveraging wildtype marginal probability, our method can predict the fitness scores of all possible single mutants in a single forward pass. In contrast, the state-of-the-art model ProteinNPT requires (D/B) forward passes to predict the fitness landscape of an assay, where D is the number of data points and B is the batch size. For predicting the A4GRB6_PSEAI_Chen_2020 assay's fitness landscape, ProteinNPT requires **58.5M GMACs with 219M parameters**, while our DePLM only requires **347.16 GMACs with 834M parameters** (792M non-trainable and 42.2M trainable). The detailed calculation process is described in Table 7.

**Diffusion steps** We investigate how the performance changes with the number of diffusion steps $T$. As illustrated in Figure 6, when $T = 0$, substituting the proposed denoising module with MLPs consistently led to lower performance, indicating the advantage of the denoising module. Interestingly, we observe that although our model can effectively learn how to denoise through the rank-based diffusion framework, the performance does not always improve with increasing $T$. The few diffusion steps in DePLM can be attributed to the following reasons: *Well-informative Initialization*: Standard diffusion models transform uninformative Gaussian noise into a complex target distribution, requiring numerous steps to capture the transformation accurately. In contrast, DePLM starts with an initial

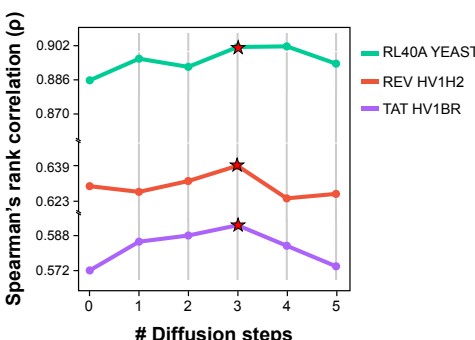

Figure 6: DePLM with varying diffusion steps. Star: the best of performance across different steps.

distribution that represents an informative protein evolutionary likelihood. This initial distribution needs only minor adjustments to reach a property-specific likelihood. Thus, DePLM requires fewer diffusion steps compared to those in standard diffusion models. *Efficient Noise Sampling*: In standard diffusion models, Gaussian noise is injected independently into each data. However, in DePLM, noise sampling considers the overall difference between the current and target distribution. A quick sorting algorithm is employed to generate a sampling pool from which we draw noises. This approach allows each step to transform the distribution more efficiently, thereby reducing the number of steps needed.

Increasing the number of diffusion steps leads to a deterioration in model performance. This decline occurs because a higher number of diffusion steps enhances the model's fitting capability, which increases the risk of overfitting to the training data as reported in [42, 5]. To further elucidate this point, we present the performance metrics of the model on both the training and test datasets in the Table 7.

Table 7: Performance of the training set and test set at different diffusion steps.

| Diffusion Step | 1 | 2 | 3 | 4 | 5 | 6 | 7 | 8 |
|---|---|---|---|---|---|---|---|---|
| Training Spearman | 0.849 | 0.864 | 0.866 | 0.881 | 0.882 | 0.883 | 0.884 | 0.886 |
| Testing Spearman | 0.694 | 0.712 | 0.716 | 0.685 | 0.587 | 0.575 | 0.576 | 0.567 |

**Structural information** When considering label-rich GB1 and Fluorescence datasets, DePLM only shows a slight improvement by incorporating the structural data. To further investigate the role of structural information, we conducted additional evaluations using the label-sparse ProteinGym assays. The results are presented in Table 8, which demonstrates a consistent enhancement in the Spearman correlation coefficient of approximately 2.7% when incorporating structures.

Table 8: Ablation study of the structural information (SI).

| Method | A0A192B1T2 | A0A247D711 | A0A2Z5U3Z0 | A4 | A4D664 | AACC1 | ACE2 |
|---|---|---|---|---|---|---|---|
| DePLM w/ SI | **0.806** | **0.565** | **0.538** | **0.741** | **0.810** | **0.654** | 0.712 |
| DePLM w/o SI | 0.777 | 0.544 | 0.494 | 0.740 | 0.799 | 0.636 | 0.707 |

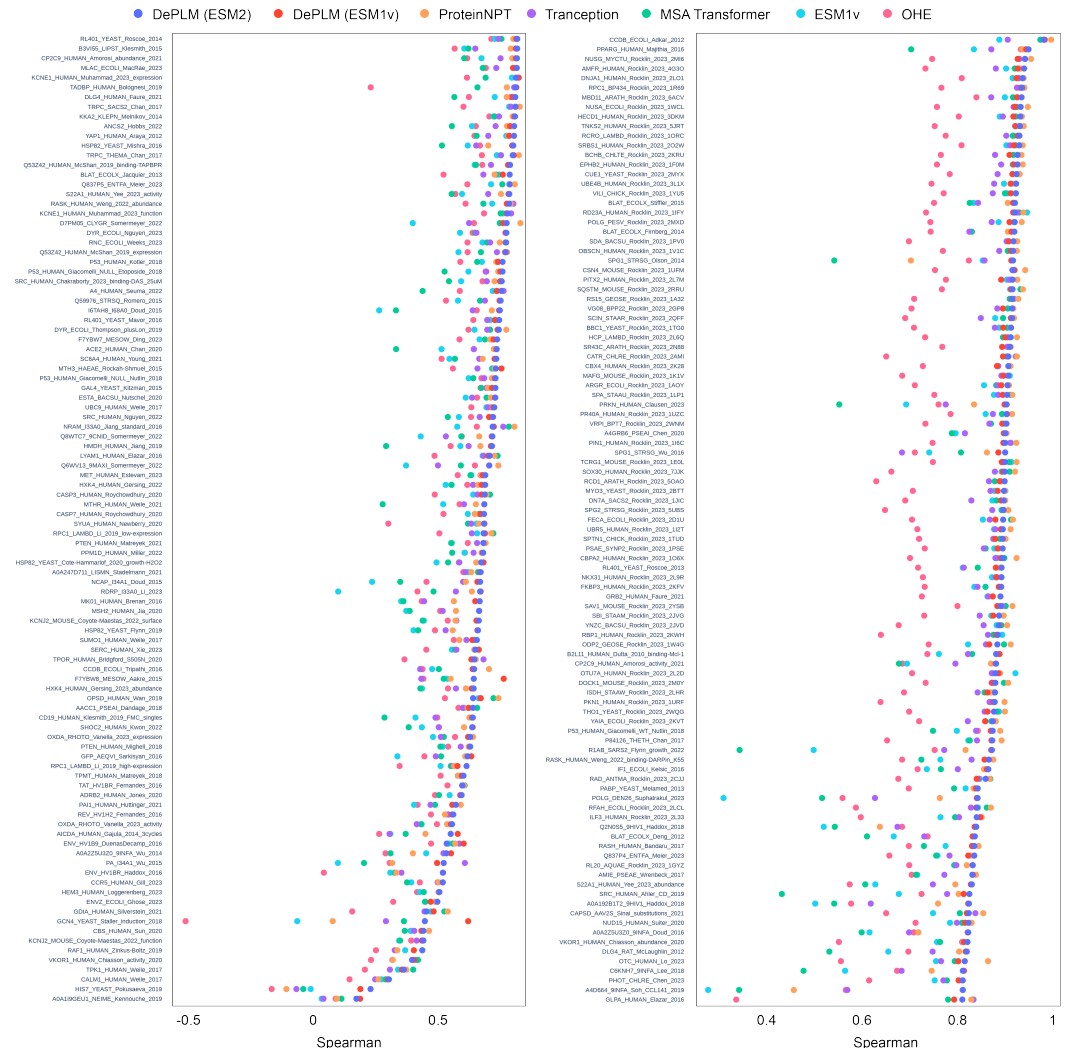

Figure 7: **Results random cross-validation scheme.** We report the DMS-level performance (measured by the Spearman's rank correlation $\rho$ between model scores and experimental measurements) of DePLM and other baselines

Table 9: Model performance on protein engineering tasks. We report mean $\pm$ standard deviation performance over random splits. The **best** results are labeled with bold.

| Model | ProteinGym | | | | |
|---|---|---|---|---|---|
| | Stability | Fitness | Expression | Binding | Activity |
| ProteinNPT | **0.904** $\pm$ 0.015 | 0.668 $\pm$ 0.035 | 0.736 $\pm$ **0.023** | 0.706 $\pm$ 0.060 | 0.680 $\pm$ 0.026 |
| DePLM (ESM2) | 0.897 $\pm$ **0.013** | **0.707** $\pm$ **0.027** | **0.742** $\pm$ 0.027 | **0.764** $\pm$ **0.041** | **0.693** $\pm$ **0.024** |

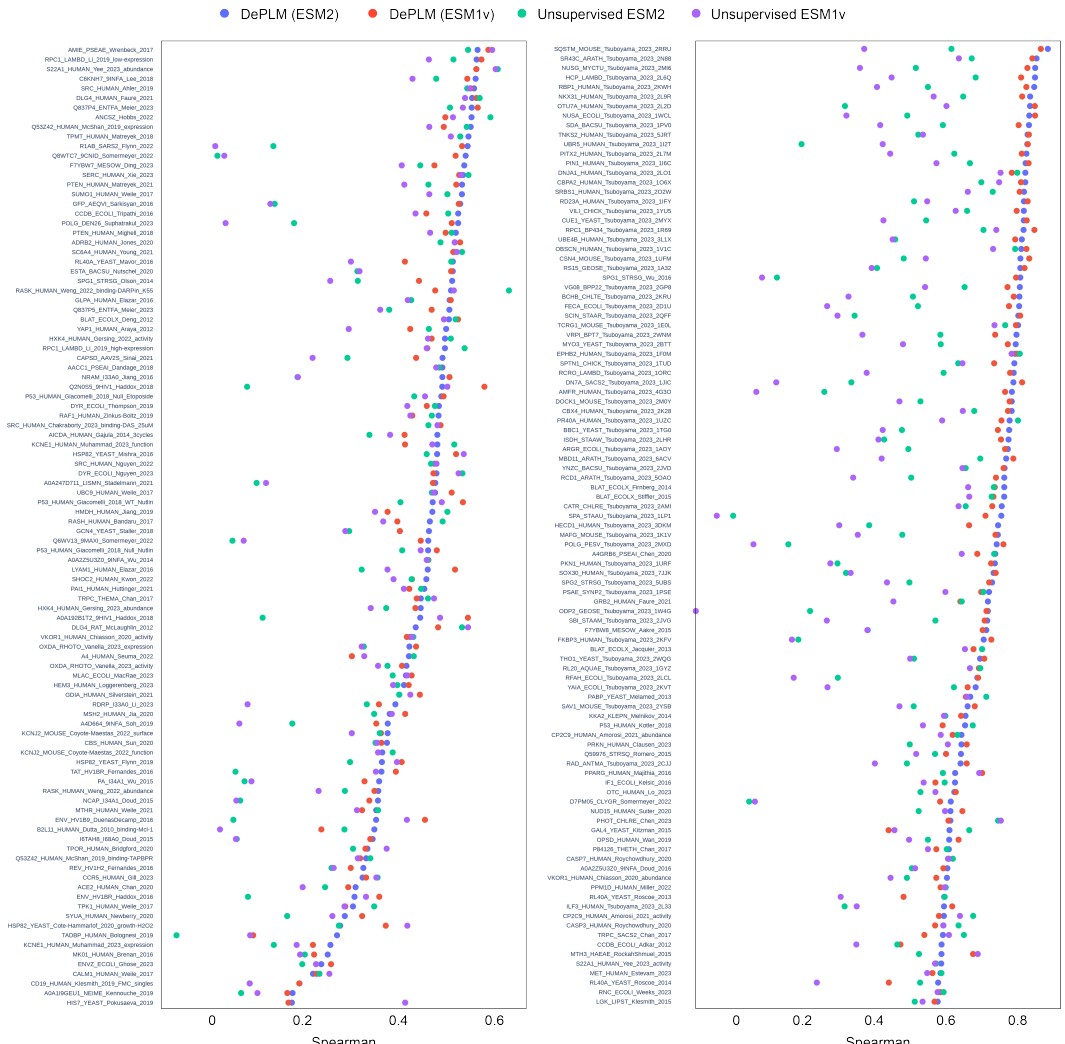

Figure 8: **Results of generalization ability.** We report the DMS-level performance (measured by the Spearman's rank correlation $\rho$ between model scores and experimental measurements) of DePLM and other baselines

