# OpenReview forum: "DePLM: Denoising Protein Language Models for Property Optimization"
_NeurIPS.cc/2024/Conference — NeurIPS 2024 poster_

### Official Review · Reviewer_FrCy · 2024-07-12

**Soundness:** 2
**Presentation:** 2
**Contribution:** 2
**Rating:** 5
**Confidence:** 4

**Summary:**

This paper proposes a new method, DePLM, for supervised fine-tuning of protein language models (PLMs) for fitness prediction tasks. DePLM uses a denoising framework with a rank correlation objective to iteratively denoise PLM likelihoods and only retain the component of the likelihood that corresponds to the fitness property of interest. The paper reports that DePLM improves on previous methods for fitness prediction on ProteinGym benchmarks and a few other single-protein benchmarks.

**Strengths:**

The paper addresses an important limitation of using PLMs for fitness prediction, namely that the overall evolutionary fitness of a protein (learned by PLMs) is a combination of many different factors, some of which are irrelevant for a particular property of interest. Denoising is an intuitively appealing approach to removing such irrelevant features.

In this approach, the paper presents some interesting, novel ideas: it proposes to use a denoising framework to improve PLM likelihoods for fitness prediction, and also proposes a novel method for fine-tuning with a rank correlation objective within that denoising framework. In their notion of denoising, a sorting algorithm creates a denoising path from rankings with low correlation to the ground truth to rankings with perfect correlation to the ground truth; this is an interesting idea.

The paper also runs some interesting experiments to test the ability of fitness prediction methods to generalize from one protein to another (Q2 in the paper), as long as the fitness properties between the proteins are somewhat similar.

**Weaknesses:**

**Major:**
1. **More baselines and ablation studies are needed to show whether DePLM is state of the art.** DePLM incorporates structural information, but in the experiments claiming that DePLM outperforms baseline methods for protein fitness prediction (Table 1), none of the other methods are given structural information. Methods that leverage structural information, such as SaProt, should be included as other relevant baselines. Similarly, in Table 2 we see that DePLM is the only method that includes evolutionary, structural, and experimental labels, but other methods that do include evolutionary and structural information could be naturally finetuned to include experimental labels. The claim that “the architecture employed by our model … enables more efficient utilization of experimental data” does not seem sufficiently supported by the shown results.
2. **The datasets used in the experiments in the paper should be described more clearly.** In particular, the experimental results for baseline methods on ProteinGym are significantly different between this paper (Table 1) and previous papers, as well as the ProteinGym leaderboard available at https://proteingym.org/. For example, ProteinNPT reports an average Spearman correlation of 0.65 on ProteinGym when performing cross-validation on random splits of the data, but in Table 1 of this paper, ProteinNPT has Spearman correlation higher than 0.65 for every subset of ProteinGym. The paper does not explain why there is this discrepancy.
3. **Describing the method under the diffusion framework is confusing and obscures what methods should be relevant comparisons to DePLM.** The theoretical basis of diffusion models relies on specifying a forward noise process that, as the time step t grows, turns complex data distributions into a known distribution that we can sample from. Learning the reverse process allows us to sample from the complex data distribution. In this paper, there is not a stochastic forward noise process, there is no noisy distribution as t -> infinity (there is a deterministic rank ordering), and there is no data distribution at t=0 that the method is attempting to sample from (again there is a deterministic rank ordering). This method seems accurately described as a denoising model, but it is confusing to me to use the diffusion framework.
4. **There are no error bars on experimental results.** It is well known that some ProteinGym datasets are harder than others, so performance must vary significantly based on the random cross-validation split, and it is unclear how many of the results are statistically significant.
5. Overall, I think that this paper presents an interesting novel method, but needs more thorough experiments with stronger baselines to show that there is a real gain in performance from their complex architecture and modeling pipeline, compared to simpler ways of integrating sequence, structure, and experimental information together.

**Minor:**
* Line 133 “addictive” -> additive
* Line 299 “Limitation” -> Limitations

**Questions:**

1. Do you have comparisons between DePLM and structure-aware models like SaProt and ProtSSN on protein fitness prediction? (Either with finetuning, or at least a simple baseline where they are used as part of a One Hot Encoding supervised method?)
2. Why are the baseline results on ProteinGym different between previously published results and what you show in Table 1?
3. Can you add error bars or other statistical analysis of the results?
4. Why does adding more denoising steps beyond 3 produce worse results? Can you determine the right number of denoising steps without ground truth labels to do hyperparameter tuning?
5. Do you apply the QuickSort algorithm with stochasticity?

**Limitations:**

The authors discuss some limitations of their work. Here are some additional points of consideration:
1. The computational complexity of their method is not directly compared with other state of the art methods, but based on the sentence, “models are trained on four Nvidia V100 32GB GPUs for 100 epochs”, it seems possible that their method is computationally much more expensive than other baselines.
2. The paper often uses “protein optimization” and “protein fitness prediction” interchangeably, but these are importantly different tasks, and the paper uses metrics traditionally associated with fitness prediction, not optimization. Spearman correlation metrics measure fitness prediction accuracy across both low and high fitness sequences, while NDCG and other similar metrics emphasize finding the best/optimal sequences out of a set. It is not clear from the experiments in the paper that you will find more optimal protein sequences using their method; generally, such experiments involve an actual optimization procedure to generate candidate sequences, and validation that these candidates are better than the candidates produced by baseline methods.

---

> ### Author Rebuttal · Authors · 2024-08-07
>
> Many thanks for the confirmation of our methodological novelty, and the constructive and valuable comments on the experiments. We have conducted addtional experiments to make the results and conclusion stronger. Here, we provide details on the comments below.
>
> >Experiments
> > 1. More baselines and ablation studies are needed to show whether DePLM is state of the art.
> > 2. There are no error bars on experimental results.
>
> Thanks for your suggestions, we have included fine-tuned versions of SaProt and ProtSSN as additional baselines in both the fitness prediction task and the generalization ability evaluation. Our revised analysis now features comprehensive statistical assessments, including mean Spearman correlation coefficients and standard deviations for each dataset, to highlight result variability. These results, detailed in the updated PDF (Tables 1 and 2), show that DePLM outperforms SaProt and ProtSSN, which incorporate evolutionary, structural, and experimental labels. Notably, DePLM achieves the smallest standard deviation on 4 out of 5 datasets, underscoring its robustness. We believe these additional results support our claim that DePLM is state-of-the-art and will incorporate them into the final draft of our manuscript.
>
> > The datasets used in the experiments in the paper should be described more clearly. Why are the baseline results on ProteinGym different between previously published results and what you show in Table 1?
>
> The discrepancy arises from differences in dataset composition. ProteinNPT in paper was evaluated on 100 DMS assays, while the leaderboard includes 216 assays, resulting in an average Spearman correlation coefficient of 0.73. Our paper focuses on a subset of 201 DMS assays due to PLM context length limitations (Appendix C.2), explaining the higher Spearman correlation observed in Table 1.
>
> >Diffusion Process
> > 1. Describing the method under the diffusion framework is confusing and obscures what methods should be relevant comparisons to DePLM.
> > 2. Do you apply the QuickSort algorithm with stochasticity?
>
> We address the concerns regarding our use of the diffusion framework with the following points:
> 1. Evolution optimizes multiple properties simultaneously, often obscuring the specific optimization objective of interest. Modeling the removal of irrelevant properties as a denoising process is reasonable. By framing it this way, we can leverage denoising models to effectively address the problem, aligning with the overall methodology and objectives of our research.
> 2. Extending the diffusion model to handle the likelihood order, deterministic at t=0 and t=\infty, represents a significant challenge and innovation in our work. We introduce a novel approach by using a sorting algorithm to identify the noise sampling space. The randomness inherent in the pivot index selection during applying the quicksort algorithm ensures that the forward process integrates the necessary stochasticity, aligning with the principles of the diffusion framework. Thus, the answer to the question about applying QuickSort with stochasticity is affirmative.
>
> > Why does adding more denoising steps beyond 3 produce worse results? Can you determine the right number of denoising steps without ground truth labels to do hyperparameter tuning?
>
> This decline occurs because a higher number of diffusion steps enhances the model's fitting capability, which also increases the risk of overfitting to the training data, as previously reported in [1, 2].
>
> Determining the optimal number of diffusion steps without ground truth labels is challenging. However, empirical evidence suggests that DePLM requires fewer denoising steps than standard diffusion models. This efficiency stems from two key advantages:
> 1. **Well-informative Initialization**: Unlike standard diffusion models that transform uninformative Gaussian noise into a complex target distribution, DePLM starts with an informative evolutionary likelihood. This requires only minor adjustments to reach a property-specific likelihood, resulting in requiring fewer diffusion steps compared to those in standard diffusion models.
> 2. **Efficient Noise Sampling**: In standard diffusion models, Gaussian noise is injected independently into each data. However, in DePLM, noise sampling considers the overall difference between the current and target distribution. A quick sorting algorithm is employed to generate a sampling pool from which we draw noises. This allows each step to transform the distribution more efficiently, thereby reducing the number of steps needed.
>
> > Clarifying the computational complexity of DePLM
>
> DePLM offers significant computational efficiency. It predicts the fitness scores of all possible single mutants in one forward pass, while ProteinNPT requires (D/B) forward passes (D for the number of mutants and B for the batch size). For the A4GRB6_PSEAI_Chen_2020 assay, DePLM requires only 347.16 GMac, compared to ProteinNPT's 58M GMac: 11,724.83 GMac per mutant * 5001 mutants. These calculations, performed using the ptflops package, highlight DePLM's efficiency. The detailed calculation process is described in Table 4 of the uploaded PDF.
>
> > Clarification of "protein optimization" and "protein fitness prediction"
>
> Thank you for highlighting the distinction between protein optimization and protein fitness prediction. We acknowledge the important differences between these two tasks, though there are strong correlations between them. In the final version of the paper, we will revise the terminology to more accurately reflect our focus, ensuring clarity and precision in our language. Additionally, we will include a discussion on the potential for future work to explore optimization procedures that generate candidate sequences, providing a pathway to validate improvements over baseline methods.
>
> [1] Improved Denoising Diffusion Probabilistic Models. Nichol et al. ICML 2021.
>
> [2] Extracting Training Data from Diffusion Models. Carlini et al. 2023.

---

> > ### Comment · Reviewer_FrCy · 2024-08-11
> > **Thank you**
> >
> > Thank you to the authors for their additional results and careful response.
> >
> > I still feel that the use of the diffusion terminology could be better motivated and explained. As I stated in the original review, I think the denoising framework is appropriate. Denoising does not require diffusion though, and it felt more confusing than illustrative to describe the addition of noise as a diffusion process, when it has a fixed end point.
> >
> > With that being said, based on the additional baseline results and the other reviewer's comments, I am willing to increase my score.

---

> > > ### Author Response · Authors · 2024-08-12
> > > **Thanks!**
> > >
> > > Thank you very much for your feedback! We are pleased that we could address your concerns.

---

### Official Review · Reviewer_ZGbQ · 2024-07-12

**Soundness:** 3
**Presentation:** 3
**Contribution:** 3
**Rating:** 7
**Confidence:** 2

**Summary:**

In this work authors tackle the problem of optimizing protein sequences towards a given property. They outline limitations of existing methods using Protein Language Models within an optimization loop to optimize property as those pLMs are not tailored towards a given property. They introduce a rank-based diffusion model to fine-tune pLMs on mutation effect prediction. This scheme can be used to optimize a wild type sequence towards a given property. They evaluate their method on four datasets and reach state-of-the-art performance. They also show that their method is able to generalize across different datasets, allowing to overcome the common problem of data scarcity in protein optimization.

**Strengths:**

**Clear Motivation**
- The paper is well-written and easy to follow. While I am not an expert on protein optimization, authors adequately contextualize their work and the motivation of the work is clear.
- The idea to adapt a diffusion-based process towards ranking is clever and relevant in the scope of protein optimization. The method is well described and derivations are correct. The method is simple to reproduce with associated pseudo-code.

**Experiments**
- The experiments are clear and demonstrate the benefit of the proposed approach compared to other baselines.
- Authors also provide interesting insights and discussion in Section 4.5 to further justify the necessity of filtering property-irrelevant information for protein optimization.
- The benefit of the ranking objective, which is the main contribution of the paper, is clearly demonstrated by the ablation study.

**Weaknesses:**

**Choice of Architecture**
- The ablation study shows very marginal improvement of using the structural information. Since this information is typically obtained through modules more expensive than sequence modules, the benefit of this part of the model is unclear.
- The detailed architecture of DePLM is not provided and some hyperparameter choices look arbitrary. For instance, the choice of 3 diffusion steps is surprising and only justified by an ablation studies on 3 small datasets.

**Questions:**

- Could you clarify and quantify the cost of training DePLM without the different modules in the ablation studies, to put in perspective with the associated performance gain ?
- Could you provide detailed choice of parameters for DePLM, notably for the different components of the denoising module ?
- The generalization experiment still shows a significant gap with training and testing on the same data source. I understand gathering new labeled data induces wet-lab costs. Would it make sense from a real-world application point of view, for a given test dataset, to combine data from other datasets and the corresponding training dataset to fine-tune DePLM ?

Some typos:
- l.80 "widetype"
- l.200 denosing

**Limitations:**

I believe authors adequately address limitations of their work.

---

> ### Author Rebuttal · Authors · 2024-08-07
>
> We appreciate Reviewer ZGbQ's constructive comments, which have significantly improved our paper. Below, we address each comment in detail.
>
> > Marginal improvement from structural information
>
> The marginal improvement observed using the structural information can be attributed to the dataset selection. For GB1 and Fluorescence, the extensive number of training mutants leads to only a slight enhancement from the inclusion of structural data. Furthermore, in our additional evaluations using the data-sparse ProteinGym assays, we observed a consistent improvement in the Spearman correlation coefficient by approximately 2.7%. The updated results are presented in Table 3 of the uploaded PDF file.
>
> > The detailed architecture of DePLM is not provided and some hyperparameter choices look arbitrary. For instance, the choice of 3 diffusion steps is surprising and only justified by an ablation studies on 3 small datasets.
>
> The choice of 3 diffusion steps in DePLM is contributed to the following reasons:
> 1. **Well-informative Initialization**: Standard diffusion models transform uninformative Gaussian noise into a complex target distribution, requiring numerous steps to capture the transformation accurately. In contrast, DePLM starts with an initial distribution that represents an informative protein evolutionary likelihood. This initial distribution needs only minor adjustments to reach a property-specific likelihood. Thus, DePLM requires fewer diffusion steps compared to those in standard diffusion models.
> 2. **Efficient Noise Sampling**: In standard diffusion models, Gaussian noise is injected independently into each data. However, in DePLM, noise sampling considers the overall difference between the current and target distribution. A quick sorting algorithm is employed to generate a sampling pool from which we draw noises. This approach allows each step to transform the distribution more efficiently, thereby reducing the number of steps needed.
>
> It is important to note that while increasing the number of diffusion steps can enhance the model's fitting ability, it also introduces a risk of overfitting as reported in [1, 2]. This results in an initial improvement in model performance, followed by a decline as the number of diffusion steps continues to rise. Therefore, we empirically determined that setting the number of steps to 3 provided the optimal trade-off. To further elucidate this point, we have included a table below that demonstrates the effects of varying the number of diffusion steps:
>
> |Assay|step 1|step 2|step 3|step 4|step 5|
> |---|---|---|---|---|---|
> |BLAT_ECOLX|0.699|0.780|0.796|0.787|0.771|
> |CALM1_HUMAN|0.252|0.338|0.339|0.338|0.308|
> |DLG4_RAT|0.852|0.858|0.853|0.851|0.839|
> |DYR_ECOLI|0.709|0.737|0.718|0.728|0.731|
>
> > Clarify and quantify the cost of training DePLM, to put in perspective with the associated performance gain.
>
> In our experimental setup, training is conducted solely with the Feature Encoder and the Denoising Block. Each training process involves one forward pass through the Protein Language Model (PLM), incurring a cost of **180.56 GMac**, and one forward pass through the Structure Encoder, incurring a cost of **77.55 GMac**. The computational overhead of the PLM is essential as it provides the evolutionary distribution as the initial state for denoising. From the analysis of the additional results, we observe that the structural encoder contributes a performance improvement of approximately 0.02.
>
> Over 100 epochs, the computational cost for the Feature Encoder amounts to **4101 GMac** (= 100 epochs * 41.01 GMACs per epoch). For the Denoising Block, the cost totals **4803 GMac** (= 100 epochs * 3 steps * 16.01 GMACs per forward). These components contribute performance improvements of 0.08 and 0.34, respectively, on the ProteinGym dataset.
>
> Overall, these designs are beneficial in enhancing performance. The additional computational cost of the different modules is minimal, making it a worthwhile trade-off for the observed performance improvement.
>
> > Could you provide detailed choice of parameters for DePLM, notably for the different components of the denoising module?
>
> For the feature encoder in DePLM, we set the sequence state dimension to 1280 and the attention head size to 32. The pairwise-residue state dimension is 32, with a matching attention head size. We apply a dropout rate of 0.2. In the denoising block, the MLP hidden dimension for converting likelihood to representation is set to 1280, using GELU activation. These parameter choices were systematically determined through extensive experimentation and cross-validation to optimize the performance of DePLM while ensuring computational efficiency. For a comprehensive description of the implementation details, please refer to the Supplementary Material (Code: archive>src>models>DePLM_module.py).
>
> > Would it make sense from a real-world application point of view, for a given test dataset, to combine data from other datasets and the corresponding training dataset to fine-tune DePLM?
>
> To explore this, we performed experiments using two different mixing strategies to construct the training dataset: (1) combining data from other datasets with 1/4 of the corresponding training data, and (2) using only 1/4 of the corresponding training data. The results of these experiments are summarized in the table below:
>
> |Dataset|Combined dataset|Only training dataset|
> |---|---|---|
> |A4GRB6_PSEAI|0.845|0.849|
> |CAPSD_AAV2S|0.619|0.584|
> |DLG4_RAT|0.722|0.717|
> |GLPA_HUMAN|0.783|0.722|
>
> The results indicate that when the corresponding training data is insufficient, incorporating data from other datasets with similar properties significantly improves model performance. We will include these findings in the final draft to emphasize the practical benefits of data integration.
>
> [1] Improved Denoising Diffusion Probabilistic Models. Nichol et al. ICML 2021.
>
> [2] Extracting Training Data from Diffusion Models. Carlini et al. 2023.

---

> > ### Comment · Reviewer_ZGbQ · 2024-08-13
> > **Answer to authors' rebuttal**
> >
> > I appreciate the authors' clarification on the ablation studies and believe these additional results will further strengthen their work. Authors also adequately addressed my other comments.
> > This confirms my initial assessment and I believe the paper should be accepted.

---

> > > ### Author Response · Authors · 2024-08-13
> > > **Thanks!**
> > >
> > > Thank you for your valuable feedback! We are glad that our responses have successfully resolved your concerns.

---

### Official Review · Reviewer_5NpU · 2024-07-13

**Soundness:** 2
**Presentation:** 3
**Contribution:** 3
**Rating:** 6
**Confidence:** 3

**Summary:**

In this work, the authors propose Denoising Protein Language Models (DePLM) to enhance protein optimization by refining evolutionary information (EI) in protein language models (PLMs). Since traditional methods struggle with considering multiple functional properties simultaneously and lack generalizability to novel proteins due to experimental condition-specific measurements. DePLM addresses these issues by denoising EI to remove irrelevant information, improving model generalization and ensuring dataset-agnostic learning. Experimental results demonstrate DePLM's superior performance in mutation effect prediction and generalization to new proteins.

**Strengths:**

1. The work develop diffusion model for protein property optimization which is a novel application.
2. The framework adapts important domain knowledge like rank-based measurement and fusion structure/sequence features.
3. The model achieves superior performance in several benchmarks.

**Weaknesses:**

1. The computational cost of the proposed method has been well discussed.

**Questions:**

1. The framework relies on a sorting algorithm to define the forward diffusion process. What is the computational cost? Will it be an obstacle for applying the proposed method to problems at scale?
2. Following the previous question, how does the computational cost (flops and # parameters) compare with baseline models?
3. DePLM only uses 3 diffusion steps, which is much less than standard diffusion model. I wonder what would happen if more steps are used? Also, what is the reason that DePLM doesn't need multiple sampling steps?
4. In Table 1, DePLM (ESM2) achieves superior performance. Does the authors by any chance have the results with ESM2 alone?
5. In Table 3, it is shown that structural information doesn't make big difference in performance. What could be the cause of this?
In Figure 3c, it seems the correlation increases with forward diffusion process. Should it be in the opposite way? Maybe I didn't fully understand it, buts when noise is added, the correlation between the optimal $\Pi^*$ and current $\Pi$ should decrease.

**Limitations:**

The authors sufficiently addressed the limitations in the submission.

---

> ### Author Rebuttal · Authors · 2024-08-07
>
> We sincerely thank Reviewer 5NpU for your insightful feedback. We have addressed your concerns below and hope our responses provide clarity:
>
> > Computational Cost
> > 1. The framework relies on a sorting algorithm to define the forward diffusion process. What is the computational cost? Will it be an obstacle for applying the proposed method to problems at scale?
> > 2. Following the previous question, how does the computational cost (flops and # parameters) compare with baseline models?
>
> We utilize the QuickSort algorithm, which has a time complexity of  $O(n\log n)$. Given the sparsity of labels, sorting the assay with the most labels in ProteinGym (\~500k mutants) takes only 1.45 seconds on 2.8GHz Quad-Core Intel Core i7, while sorting the assay with the median number of labels (\~5k mutants) takes just 0.0056 seconds.
>
> By leveraging wildtype marginal probability, our method can predict the fitness scores of all possible single mutants in a single forward pass. In contrast, the state-of-the-art model ProteinNPT requires (D/B) forward passes to predict the fitness landscape of an assay, where D is the number of data points and B is the batch size. For predicting the A4GRB6_PSEAI_Chen_2020 assay's fitness landscape, ProteinNPT requires **58.5M GMACs with 219M parameters**, while our DePLM only requires **347.16 GMACs with 834M parameters** (792M non-trainable and 42.2M trainable). The detailed calculation process is described in Table 4 of the uploaded PDF.
>
> Overall, DePLM proves to be an effective and efficient method for predicting protein fitness landscapes and the inclusion of the ranking algorithm does not hinder scalability compared to baseline models.
>
> > Diffusion Step
> > 1. DePLM only uses 3 diffusion steps, which is much less than standard diffusion model. I wonder what would happen if more steps are used? Also, what is the reason that DePLM doesn't need multiple sampling steps?
> > 2. In Figure 3c, it seems the correlation increases with forward diffusion process. Should it be in the opposite way? Maybe I didn't fully understand it, buts when noise is added, the correlation between the optimal $\Pi^{\star}$ and current $\Pi$ should decrease.
>
> The few diffusion steps in DePLM can be attributed to the following reasons:
> 1. **Well-informative Initialization**: Standard diffusion models transform uninformative Gaussian noise into a complex target distribution, requiring numerous steps to capture the transformation accurately. In contrast, DePLM starts with an initial distribution that represents an informative protein evolutionary likelihood. This initial distribution needs only minor adjustments to reach a property-specific likelihood. Thus, DePLM requires fewer diffusion steps compared to those in standard diffusion models.
> 2. **Efficient Noise Sampling**: In standard diffusion models, Gaussian noise is injected independently into each data. However, in DePLM, noise sampling considers the overall difference between the current and target distribution. A quick sorting algorithm is employed to generate a sampling pool from which we draw noises. This approach allows each step to transform the distribution more efficiently, thereby reducing the number of steps needed.
>
> Increasing the number of diffusion steps leads to a deterioration in model performance, as illustrated in Figure 8. This decline occurs because a higher number of diffusion steps enhances the model's fitting capability, which increases the risk of overfitting to the training data as reported in [1, 2]. To further elucidate this point, we present the performance metrics of the model on both the training and test datasets in the table below:
>
> | Step   | 1     | 2     | 3     | 4     | 5     | 6     | 7     | 8     |
> | --- | ----- | ----- | ----- | ----- | ----- | ----- | ----- | ----- |
> | Training Spearman | 0.849 | 0.864 | 0.866 | 0.881 | 0.882 | 0.883 | 0.884 | 0.886 |
> | Testing Spearman  | 0.694 | 0.712 | 0.716 | 0.685 | 0.587 | 0.575 | 0.576 | 0.567 |
>
> Figure 3c illustrates the relationship between the Spearman coefficient of evolution likelihood and intermediate rank variables as a function of the number of forward steps. The caption might be a bit confusing but we will correct it the final version.
>
> > Model Architecture
> > 1. In Table 1, DePLM (ESM2) achieves superior performance. Does the authors by any chance have the results with ESM2 alone?
> > 2. In Table 3, it is shown that structural information doesn't make big difference in performance. What could be the cause of this?
>
> The PLM-based results reported in Table 1 are sourced from the ProteinGym Leaderboard [https://proteingym.org/](https://proteingym.org/), which does not include ESM2-based results. Here, we report ESM2 results obtained by our own experiments and will include these in the final draft.
>
>
> | ProteinGym   | Stability | Fitness | Expression | Binding | Activity |
> | ------------ | --------- | ------- | ---------- | ------- | -------- |
> | ESM2         | 0.882     | 0.563   | 0.645      | 0.587   | 0.576    |
> | DePLM (ESM2) | 0.897     | 0.707   | 0.742      | 0.764   | 0.693    |
>
> Regarding the second question, the minimal performance difference observed with structural information can be attributed to the dataset selection. When considering label-richGB1 and Fluorescence datasets, DePLM only shows a slight improvement by incorporating the structural data. To further investigate the role of structural information, we conducted additional evaluations using the label-sparse ProteinGym assays. The updated results are presented in Table 3 of the uploaded PDF file, which demonstrates a consistent enhancement in the Spearman correlation coefficient of approximately **2.7%** when incorporating structures.
>
> [1] Improved Denoising Diffusion Probabilistic Models. Nichol et al. ICML 2021.
>
> [2] Extracting Training Data from Diffusion Models. Carlini et al. 2023.

---

> > ### Comment · Reviewer_5NpU · 2024-08-11
> > **Response to rebuttal**
> >
> > I thank the authors for the detailed response which have addressed most of my concerns. I still positive about this submission.

---

> > > ### Author Response · Authors · 2024-08-12
> > > **Thanks!**
> > >
> > > Thank you immensely for your feedback! We are gratified to know that we have successfully addressed your concerns.

---

### Author Rebuttal · Authors · 2024-08-07

We sincerely appreciate the reviewers' thorough and constructive feedback on our manuscript. In response, we have conducted a series of additional experiments and analyses to address your concerns and strengthen our paper. Below, we provide an overview of the new results included in the uploaded PDF:

1. **Additional Baselines**: In Tables 1 and 2, we have supplemented the results with fine-tuned ProtSSN and SaProt models on the protein fitness prediction task (Q1) and the generalization ability evaluation (Q2). In addition to the average Spearman coefficient, we also include the average standard deviation. We observe that DePLM outperforms these models, which consider sequential, structural, and experimental signals simultaneously, demonstrating the superiority of the proposed model architecture.
2. **Extended Ablations**: In Tables 3, we have included more datasets to verify the necessity of incorporating structural information (Q3). The results show that introducing structural information consistently and significantly improves the model's performance.
3. **Computational Cost Analysis**: In Table 4, we use the A4GRB6_PSEAI_Chen_2020 assay as a case study to compare the computational cost and parameter count of DePLM and ProteinNPT in both training and inference phases. The results indicate that our proposed method is much more efficient.

In the following sections, we present a detailed point-by-point response to the questions raised.

---

### Decision · Program_Chairs · 2024-09-25

**Decision:**

Accept (poster)

**Comment:**

The authors propose a new fine-tuning technique for protein language models, using a rank-based diffusion model. The reviewers highlighted that the paper is well written, that it contains interesting, novel ideas, and that the importance of the methodological advances is backed up by empirical results. Some concerns were raised regarding the computational efficiency of the method, the ablation studies, and the number of baselines, but the reviewers all stated that their concerns were (mostly) addressed by the rebuttal.  In conclusion, the paper presents several novel ideas, and obtains results superior to the current state-of-the art.